# The Risk of Invasive Pneumococcal Disease Differs between Risk Groups in Norway Following Widespread Use of the 13-Valent Pneumococcal Vaccine in Children

**DOI:** 10.3390/microorganisms9081774

**Published:** 2021-08-20

**Authors:** Brita Askeland Winje, Didrik Frimann Vestrheim, Richard Aubrey White, Anneke Steens

**Affiliations:** Division of Infection Control and Environmental Health, Norwegian Institute of Public Health, P.O. Box 222, Skøyen, 0213 Oslo, Norway; DidrikFrimann.Vestrheim@fhi.no (D.F.V.); RichardAubrey.White@fhi.no (R.A.W.); Anneke.Steens@fhi.no (A.S.)

**Keywords:** *Streptococcus pneumoniae*, invasive pneumococcal disease, comorbidities, pneumococcal vaccines, incidence

## Abstract

The elderly and adults with medical risk conditions remain at high risk of invasive pneumococcal disease (IPD), highlighting the importance of adequate preventive efforts. In an observational population-based study in Norway (pop ≥ 5 years, 2009–2017) covering six years post-PCV13 implementation, we explored the incidence and risk of IPD associated with age and comorbidities. We obtained the data on 5535 IPD cases from the Norwegian Surveillance System for Communicable Diseases and the population data from Statistics Norway. To define comorbidities, we obtained ICD-10 codes from the Norwegian Patient Registry for the cases and the Norwegian population. The average annual decrease in PCV13 IPD incidence was significant in all risk groups and decreased post-PCV13 introduction by 16–20% per risk group, implying a nondifferential indirect protection from the childhood vaccination. The IPD incidence remained high in the medical risk groups. The relative importance of medical risk conditions was 2.8 to 6 times higher in those aged 5–64 versus ≥65 years for all types of IPD, since age itself is a risk factor for IPD. In groups without medical risk, the risk of IPD was eight times higher in those aged ≥65 compared to those 5–64 years (RR 8.3 (95% CI 7.3–9.5)). Our results underscore the need for age- and risk-group-based prevention strategies.

## 1. Introduction

Vaccination with pneumococcal conjugate vaccines (PCVs) in children has had a profound effect on the epidemiology of invasive pneumococcal disease (IPD) of both targeted and nontargeted age groups [1,2]. Although the IPD incidence has decreased in the adult population, the elderly and adults with underlying medical conditions are carrying an increased proportion of the remaining IPD burden in several settings [3,4,5,6,7]. These high-risk groups also have an increased risk of IPD-associated mortality, highlighting the importance of adequate preventive efforts by vaccination [8].

An important first step to inform vaccine recommendations is to identify the population-based risk of IPD in the elderly and medical risk groups. Although some previous studies have reported on the magnitude of the increased risk compared with the general population [5,9,10,11], they are limited by the data being from before 2013 [5,11,12], that the risk conditions are only presented in broad compound groups [9], or that they report from non-European settings [10]. This renews the interest in exploring risk groups for IPD and the influence of various underlying medical conditions on the IPD epidemiology.

Two complementary European multicentre studies have studied the impact of higher-valent PCVs used in the childhood vaccination programme on the incidence of IPD in children and adults under 65 years (*Streptococcus pneumoniae* Invasive Disease Network; SpIDnet [13]) and in the population aged 65 years or older (Integrated Monitoring of Vaccines in Europe; I-MOVE [14]). The studies showed that the indirect effects of childhood vaccination had led in five years to an overall, nonsignificant decrease of 14% (95% CI −4%, 30%) in IPD in the elderly [15].

The 13-valent conjugate vaccine (PCV13) and the 23-valent polysaccharide vaccine (PPV23) are currently licensed for use in adults [16]. Due to its broader serotype coverage, PPV23 is currently the recommended vaccine for medical risk groups and the elderly in Norway and only a few, selected high-risk groups are recommended PCV13 in series with PPV23 [17]. The uptake of pneumococcal vaccination has been suboptimal in Norway, at around 15%. New pneumococcal conjugate vaccines offering protection against additional serotypes (PCV15 and PCV20) are under development and will be available for adults in the near future [18,19].

In this study, we extended the Norwegian part of the European studies to 2009–2017, including the six years post-PCV13 introduction to explore the impact and serotype distribution of IPD among medical risk groups and the elderly in a mature PCV13 program, and we were able to calculate incidence rates through inclusion of data on the size of the risk groups in the population. In order to inform vaccine recommendations, we assessed whether the indirect effects from PCV13 childhood vaccination have been the same in groups with and without medical risk conditions. We therefore calculated the age group-, medical risk group- and vaccine type-specific incidence rates of IPD in the population 5 years or older in Norway in 2009 to 2017 and estimated the risk of IPD in these groups. To explore the preventive potential of different vaccines, we studied vaccine-specific serotype distributions in the medical risk groups.

## 2. Materials and Methods

This is an observational retrospective population-based study covering 2009–2017. All Norwegian inhabitants 5 years or older were considered the source population. All data sources for the cases are summarised in Appendix A. We obtained data on IPD cases from the Norwegian Surveillance System for Communicable Diseases (MSIS). It is mandatory to notify IPD to MSIS and due to well-established routines in diagnostic laboratories the sensitivity of laboratory notifications is assumed to be very high. IPD isolates are routinely sent to the reference laboratory at the Norwegian Institute of Public Health for surveillance. All isolates were serotyped using the Quellung reaction. Vaccination history of the cases was available for cases with an index date in 2015–2017 through the Norwegian Immunisation Registry SYSVAK or through contact with the physician of the case.

We used data from the Norwegian Patient Register (NPR) [20] on hospital discharge diagnoses (International Statistical Classification of Diseases and Related Health Problems —10th Revision codes [ICD-10 codes]) to define underlying medical risk conditions. For IPD cases, we linked NPR data to the case-based data using a unique personal identification number available for all inhabitants. For the Norwegian population, we obtained annual age-stratified aggregated data on the size of the medical risk groups from NPR. The assignment of the population to risk conditions was consistent with the definition for the cases (see below). Data on the annual size of the age groups in the population were obtained from Statistics Norway [21].

We defined different medical risk conditions using ICD-10 codes (Appendix A). The selected risk conditions are, in large part, consistent with other European cohorts [13,14]. We included ICD-10 codes registered within the last two years before the index date. The index date for cases was the date of IPD diagnosis, for the population we used July 1st of the corresponding years. In Norway, hospital discharge data are only available for linkage with other sources since 2008. For 2008 and 2009, we therefore included all available data from January 2008 up to the index date. Since one person may have several risk conditions, these were not mutually exclusive. However, we classified risk conditions into (compound) risk strata: high risk (immunodeficiency, HIV, asplenia, chronic kidney disease, nephrotic syndrome, haematologic or generalised malignancies or transplantation), medium risk (CSF leakage, cochlear implant, chronic heart disease, chronic lung disease, diabetes mellitus, chronic liver disease, other malignancies, alcoholism *and* no high-risk condition), or no risk (none of the listed risk conditions). These three risk strata were mutually exclusive, with the highest risk condition determining the risk stratum.

### 2.1. Definitions

IPD re-admission less than 30 days apart with the same serotype was considered as the same episode and was not included in the analyses. We categorised the isolates by vaccine type; PCV13 (1, 3, 4, 5, 6A, 6B, 7F, 9V, 14, 18C, 19A, 19F and 23F), non-PCV13 (serotypes not included in PCV13), and serotypes included in PPV23 (PCV13 serotypes except 6A and serotypes 2, 8, 9N, 10A, 11A, 12F, 15B, 17F, 20, 22F and 33F). In addition, we classified serotypes according to two new conjugate vaccines which will be available in the near future; PCV15 (PCV13 serotypes + 22F and 33F) and PCV20 (PCV15 + 8, 10A, 11A, 12F, and 15B). Serotype information was missing for 135 cases (2.4%). For all years, the proportion of IPD cases with serotype information available was >96%. In analyses at vaccine-type level, we corrected for missing serotype information proportionally to the distribution of vaccine-types among those with available serotype information.

Having been vaccinated was defined as having a date of vaccination (with PCV or PPV23) more than 14 days but no longer than 10 years before the index date. In Norway, revaccination is recommended every 10th year.

### 2.2. Changes in IPD Incidence Rates and Estimates of Indirect Effects

The risk profile and characteristics of IPD cases are presented descriptively. Risk group-, age group-, and vaccine type-specific incidence rates of IPD were calculated for 2009 to 2017 by dividing the number of cases in the risk groups by the size of the corresponding groups in the Norwegian population and are presented per 100,000 inhabitants. Poisson regression was used to estimate the average yearly change in incidence rates since the switch to PCV13 (2011–2017) and was expressed as incidence rate ratios (IRR) with 95% confidence intervals (95% CI). Year 2011, the year of PCV13 introduction, was used as baseline-year. The IRRs present a relative measure of the indirect effects of vaccination.

### 2.3. The Relative Effects of Age and Medical Conditions on the Risk of IPD

We determined risk ratios (RR) for (vaccine-type) IPD stratified by age (5–64/65+). The RR was calculated as the ratio between the risk for IPD in the medical risk groups compared to the risk for IPD among those with no medical risk condition.

Data were analysed in StataSE 16.0 and excel 2016.

## 3. Results

In total, 5535 IPD cases aged 5 years or older were reported to MSIS over the nine-year period from 2009 to 2017; 18% (n, 970) were 5–49, 26% (n, 1448) were 50–64, and 56% (n, 3117) were ≥65 years of age. The median ages (interquartile range) within these age groups were 38 (5–48), 59 (50–64) and 77 (65–98) years, respectively. Note that in the Norwegian population, 64% of individuals were aged 5–49, 19% 50–64 and 17% ≥ 65 years.

Overall, 43% (n, 2402) of IPD cases had at least one medical risk condition registered within the last two years prior to the IPD diagnosis, compared to 12% in the Norwegian population. In total, 23% of the cases (n, 1267) had one risk condition, 14% (n, 750) had two risk conditions and 7% (n, 385) had >two risk conditions. High-risk conditions were present in 19% (n, 1027) of the IPD cases compared to 2% of the Norwegian population. The proportion with medical risk conditions was highest in the oldest age group (Table 1). Other malignancies, chronic lung disease, chronic heart disease and diabetes were the most frequent underlying conditions, both for cases and for the population.

### 3.1. Changes in Age Group-, Vaccine Type- and Medical Risk Group-Specific IPD Incidence Rates over Time

The overall IPD incidence rate in Norway was low; ~16/100,000 in the years before PCV13 use (2009–2010) and ~11/100,000 in the years 2014–2017 (stippled line, Figure 1A), following a decrease in vaccine-type IPD after the introduction of PCV7 and PCV13 in children (Figure 1B; indirect protection). IPD caused by PCV13 serotypes decreased by 73% from 2009/2010 to 2017. This was mainly due to the 68% decrease in IPD caused by the six additional serotypes that are in PCV13 but not PCV7. The decrease in PCV13 IPD continued for the whole study period for those < 65 years. In those ≥ 65 years, the decrease was substantial and without delay in the first years after PCV13 introduction, but the incidence rate increased slightly from 2015. The overall decrease in IPD was counteracted by an increase in non-PCV13 IPD (replacement disease; Figure 1C). In those aged 5 to 64 years, the incidence rate of non-PCV13 IPD increased from 2009/2010 to 2017 by 16% to 4.1/100,000. In those ≥ 65 years, non-PCV13 IPD increased by 15% to 29.3/100,000. The incidence rate of PPV23 IPD has decreased during the study period (Figure 1D). However, there is a considerable remaining disease burden of PPV23 serotype IPD in the oldest age group, in which many may be prevented through vaccination.

Figure 2A–D shows risk group-specific IPD incidence rates stratified by those aged 5–64 and ≥65 years and vaccine serotypes. The time trends in the risk groups were quite similar between the age strata while the IPD incidence rates were highest for those ≥65 years for all risk strata and for the different vaccine types. PCV13 IPD incidence rate decreased to <16/100,000 in all groups (Figure 2B). The incidence rate of non-PCV13 IPD was substantially higher in those with high-risk medical conditions compared to lower risk groups for both age strata (Figure 2C). The incidence rate of PPV23 IPD remained above 60/100,000 in the high-risk group (Figure 2D). It is noteworthy that the 65+ with no medical risk condition had a higher non-PCV13 IPD incidence rate compared to the medium risk group aged 5–64 years.

The average annual decrease in PCV13 IPD incidence rate was significant in all medical risk groups and decreased on average by 19% (95% CI 17–22) yearly, ranging from 16–20% between groups (Figure 3A, overlapping confidence intervals), implying a nondifferential indirect effect from childhood vaccination after PCV13.

Non-PCV13 IPD increased in all groups, on average by 4% (95% CI 2–6) yearly, ranging from 2–4% between groups with overlapping confidence intervals (Figure 3B). When zooming in on the separate medical risk conditions, numbers become small and thereby causing wide 95% CIs (Figure 3C,D), but the pattern over time was similar as described above.

### 3.2. Relative Risk of IPD in Medical Risk Groups Compared to Those with No Risk by Age Group

We included the 1622 IPD cases notified in 2015 to 2017 to estimate the RR of IPD. As expected, the RR was higher in all medical risk groups compared to the no risk group and highest for the high-risk conditions (Table 2). The relative importance of medical risk conditions was consistently lower in those ≥65 compared to those aged 5–64 years, since age itself is a risk factor for IPD. The RR in those ≥65 years with no risk compared to those aged 5–64 years with no risk was 8.3 (95% CI 7.3–9.5) for all IPD, 5.9 (95% CI 4.6–7.5) for PCV13 IPD, 9.7 (95% CI 8.2–11.4) for non-PCV13 IPD and 7.3 (95% CI 6.2–8.5) for PPV23 IPD.

The RR of all types of IPD in the high-risk group compared with the no risk group was 25.5 (20.6 to 31.7) and 4.2 (3.6 to 5.0) for those aged 5–64 and ≥65 years, respectively. Overall, the RR of IPD in medical risk groups compared to the no risk group was 2.8 to 6 times higher in those aged 5–64 years compared with those ≥ 65 for all IPD, 2.8 to 5 times higher for PCV13 IPD and 2.7 to 5.7 times higher for PPV23 IPD. Haematological and general malignancies were the two risk conditions with the highest relative importance in the group 5–64 years compared with those 65 years or older. The risk of IPD was higher for those with two compared to one medical risk condition, irrespective of age. Surprisingly, the risk of IPD in those with more than two medical risk conditions was lower than for those with two risk conditions.

### 3.3. Serotype-Specific IPD in Medical Risk Groups 2015–2017

The percentage of IPD cases caused by PCV13 serotypes was low (range 19–29%) overall, limiting the potential for protection from PCV13 vaccination (Table 3). Note that 1% of PCV13 cases and 9% of PPV23 cases had been vaccinated within the last 10 years prior to the index date, thereby being breakthrough infections. As expected, the percentage of cases caused by serotypes covered by current (PPV23: range 59–76%) and new vaccines (range PCV15: 31–49% and PCV20: 46–65%) increased with the number of serotypes included in the vaccine. However, the percentage with potentially vaccine preventable IPD was lowest in the high-risk group and in individuals with multiple risk conditions (Table 3).

In 2015–2017, the most frequently reported serotypes among the 1593 IPD cases with serotype information available were 22F (n = 294, 18%), 3 (n = 163, 10%), 8 (n = 91, 6%), 9N (n = 88, 6%), 7F (n = 85, 5%), 33F (n = 70, 4%) and 19A (n = 59, 4%). Serotype 22F and 3 were the two most common serotypes across all risk strata; serotype 22F accounted for 20%, 21% and 12% in the no-, medium- and high-risk groups, respectively, and for serotype 3 for 12%, 8% and 8%. Serotype 3 accounted for about 40% of the PCV13 IPD cases across all risk strata (range 40–42%).

## 4. Discussion

The switch from PCV7 to PCV13 in the childhood vaccination programme in Norway has led to nondifferential indirect protection in all medical risk groups, but medical risk groups remain at a substantially higher risk for IPD than those without. Age is an important risk factor for IPD with eight times higher risk in those aged ≥65 years compared to those 5–64 years in groups without medical risk conditions. Because of this age effect, the impact of having medical risk conditions on the risk for IPD is much larger for younger age groups. The PCV13 IPD incidence rate has become low in all groups, limiting the potential for prevention through PCV13 vaccination, whereas PPV23 serotypes still account for a substantial proportion of IPD in groups targeted for pneumococcal vaccination. Extended-valent (15- and 20-valent) conjugate vaccines will be available in the near future and may change vaccine recommendations [18,19]. Unfortunately, since IPD caused by nonvaccine serotypes are more common among those with high-risk conditions, the preventive potential of the different vaccines is lower in these groups that need it most.

### 4.1. Incidence Rate by Age and Vaccine-Type IPD

Despite the substantial reduction in PCV13 IPD in Norway, the overall IPD reduction in the population 5 years and older was more modest, due to a 20% increase in non-PCV13 IPD. Increase of nonvaccine serotype IPD in adults 65+ have negated reductions in vaccine-type disease in several European [15] and non-European countries [22,23,24]. Ladhani et al., reported a 77% increase in non-PCV13 IPD IR in those 65 years and older in England and Wales in the period 2008/2010 to 2016/2017, which is considerably higher than the 15% increase in incidence rates reported in the current study. Still, the non-PCV13 IPD IR remains higher in Norway compared to England and Wales (29.3 versus 22.7 per 100,000) [25]. In contrast, the non-PCV13 incidence rate in the USA has remained stable after introduction of PCV13 [10]. Possible explanations are discussed in detail by Lewnard et al. and include secular, biological, methodological and demographic factors [26]. Although, serotype replacement has been more modest in Norway than in many other countries, the pattern of replacement in Norway is more similar to other European settings, than to the USA. The Nordic countries and the Netherlands had substantially higher prevaccine IPD incidence rates than most European countries [15]. The reduction in PCV13 IPD in adults in Norway followed the changes in children with some delay and then stabilised from 2015 and onwards. It is postulated that the maximum benefit of the childhood PCV10/13 vaccination may have been reached, and that further decreases in overall and PCV13 IPD are unlikely in mature PCV13 settings with high uptake [27,28].

### 4.2. Indirect Effect in Groups with and without Medical Risk Factors

We found a nondifferential indirect effect from childhood vaccination in medical risk groups compared to the no-risk group. The majority of previous studies reported nondifferential effects [4,5,6,10,29] in line with the current study, whereas some reported smaller indirect effects for risk groups and immunosuppressed individuals [3,4], although a benefit was seen in case-fatality rates [4]. An earlier Norwegian study (2005–2014) focusing on iatrogenic immunosuppression, concluded that the entire population benefited from childhood vaccination [30]. The 95% CIs for the separate medical risk conditions were wide due to low case numbers. However, the overall patterns were similar as the grouped estimates, with some variations.

### 4.3. Risk of IPD in Medical Risk Groups

Previous studies have reported the prevalence of high-risk conditions among IPD cases in the range of 15–36% with the majority being close to the 19% reported in the current study [4,5,6,11]. In contrast, the prevalence of medium risk conditions is reported to be higher (range 55–59%) in other studies [4,6,11], compared to the 25% presented here, since we assigned individuals to mutually exclusive risk strata with the highest risk condition determining the risk group. We included ICD-10 codes registered within the last 2 years prior to the IPD episode to define risk conditions. This information was not available from other studies. A shorter period to define the risk conditions may affect the prevalence of risk conditions as well as the severity of them. The percentage of IPD cases with underlying comorbidities changes over time [9,10], making comparisons between previous and more recent studies difficult. The number of cases in specific high-risk groups in our data were too small to do meaningful calculations, but our results are overall consistent with other studies reporting a marked increased risk of IPD among those with certain clinical conditions, such as haematological cancer, generalised malignancies and solid organ transplants [30,31,32].

While PCV13 IPD incidence rates have decreased to low levels across all risk strata, the non-PCV13 IPD incidence rates remained substantially higher in individuals with high-risk conditions in both age groups. Age itself is associated with increased risk of IPD due to immunosenescence, the age-associated alterations of the immune system leading to increased susceptibility to infectious pathogens in older adults [33]. We found that the overall IPD IR in the elderly ≥65 years or older, even in the absence of medical risk conditions, outweighed the IR in groups < 65 with medium-risk conditions. This result was further strengthened by the lower relative importance of medical risk conditions for IPD in those ≥ 65, compared to those 5–64 years, implying that age alone contributes substantially to a higher risk of IPD in older adults. Similar results have been reported previously [5,11,34]. Although the risk estimates differed somewhat between the studies, the relative difference in those </≥ 65 years were similar. The role of ageing on the risk of infections and prevention strategies has become even more clear during the COVID-19 pandemic. Strategies to protect the old and frail populations from infectious diseases need careful consideration when planning public health interventions.

### 4.4. Strengths and Weaknesses

The strengths of this study include the well-established population-based health registries, the opportunity to link registry data at an individual level through a personal ID number and the high coverage of serotype information. Having similar data on the prevalence of risk conditions in both cases and the total population allowed us to calculate IPD incidence rates associated with specific risk conditions, to look at incidence rate ratios and to calculate incidence rates in risk groups relative to the groups without risk conditions. Further, we were able to assess the effect of having several comorbidities. Relatively few studies report prevalence estimates of risk conditions in the general population and estimates are based on the extrapolation of data from other sources [5,9,10,11].

Limitations include the fact that we have no information on disease severity, risk behaviours, such as excessive alcohol consumption and smoking and no information on iatrogenic immunosuppression. The latter may cause misclassification, i.e., individuals on immunosuppressive therapy with no registration in the NPR within the last two years may be classified as having no risk. In a sensitivity analysis performed on Norwegian data for 2009–2014 where NPR data and data from the prescription register were linked, we found that of those defined as not having a medical risk condition, 4.9% used immunosuppressive drugs that increase the risk for IPD (Appendix A). This would lead to an underestimation of IPD in medical risk groups and an overestimation of IPD in the no risk group. We may also have underestimated less severe risk conditions, if the persons were only treated by their general practitioner. Further, individuals may have several risk conditions, and the estimated IPD incidence rate for individual comorbidities may be affected by the presence of additional comorbid conditions.

## 5. Conclusions

All age and medical risk groups have benefitted from indirect protection, although the individuals with underlying medical conditions remain at higher risk of IPD, specifically those with high-risk conditions. The relative importance of medical risk was substantially lower in the oldest age group, underpinning the need for age- and risk-group-based prevention strategies. The low PCV13 IPD in all groups limits the potential for prevention through PCV13 vaccination whereas PPV23 serotypes still account for a substantial proportion of IPD in groups targeted for pneumococcal vaccination, though to a lesser extent for those who need it most. Extended-valent (15- and 20-valent) conjugate vaccines will be available in the near future and may change vaccine recommendations.

## Figures and Tables

**Figure 1 microorganisms-09-01774-f001:**
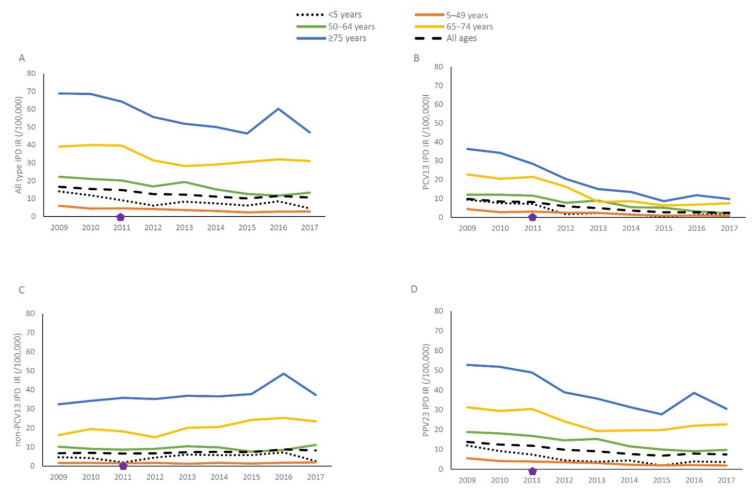
IPD incidence rate by age group in Norway 2009 to 2017. (**A**) All types of IPD, (**B**) PCV13 IPD, (**C**) non-PCV13 IPD, (**D**) PPV23 IPD, Diamonds indicate the year of PCV13 vaccine introduction (2011: PCV13). All ages and age group <5 years (stippled and dotted lines) are included as references. All types of IPD includes IPD caused by any serotype. PCV13 IPD includes serotypes 1, 3, 4, 5, 6A, 6B, 7F, 9V, 14, 18C, 19A, 19F and 23F; non-PCV13 IPD includes serotypes not included in PCV13; PPV23 IPD includes serotypes included in PCV13 except serotype 6A and serotypes 2, 8, 9N, 10A, 11A, 12F, 15B, 17F, 20, 22F and 33F.

**Figure 2 microorganisms-09-01774-f002:**
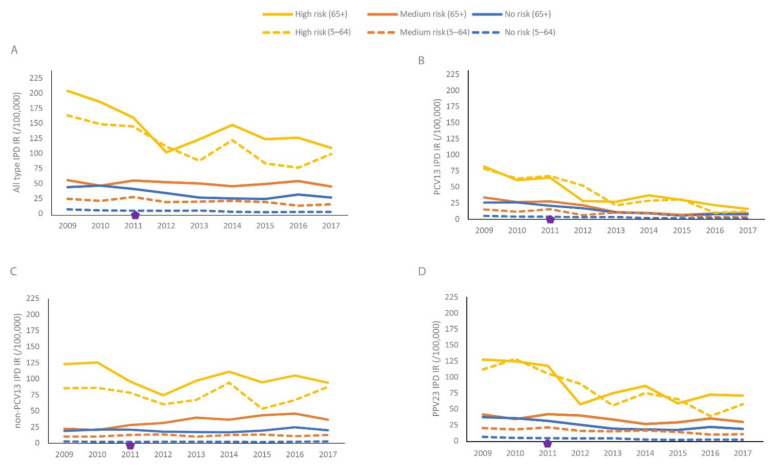
IPD incidence rate by medical risk strata, by age (5–64 and ≥65 years). (**A**) All types of IPD, (**B**) PCV13 IPD, (**C**) non-PCV13 IPD, (**D**) PPV23 IPD. Diamonds indicate the year of PCV13 vaccine introduction (2011: PCV13). The three risk groups are mutually exclusive, with the highest risk condition determining the risk group. All types of IPD includes IPD caused by any serotype. PCV13 IPD includes serotypes 1, 3, 4, 5, 6A, 6B, 7F, 9V, 14, 18C, 19A, 19F and 23F; non-PCV13 IPD includes serotypes not included in PCV13; PPV23 IPD includes serotypes included in PCV13 except serotype 6A and serotypes 2, 8, 9N, 10A, 11A, 12F, 15B, 17F, 20, 22F and 33F.

**Figure 3 microorganisms-09-01774-f003:**
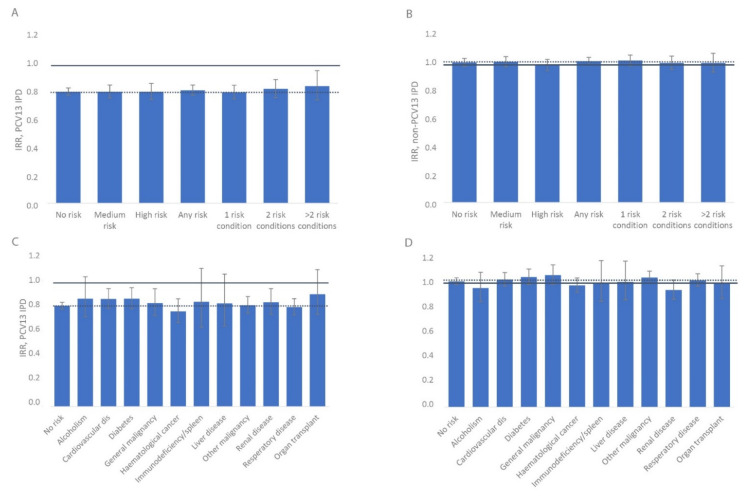
Average annual change in IPD incidence rate, by medical risk in the population aged 5 years or older, Norway, 2011 to 2017. (**A**) PCV13 IPD, (**B**) non-PCV13 IPD, (**C**) PCV13 IPD, (**D**) non-PCV13 IPD. PCV13 serotypes are 4, 6B, 9V, 14, 18C, 19F, 23F, 1, 3, 5, 6A, 7F and 19A; non-PCV13 serotypes are all serotypes not included in PCV13. The solid line indicates 1 for significance. The stippled line reflects the level of the no risk group.

**Table 1 microorganisms-09-01774-t001:** Characteristics of the study population. Number (%) of medical risk conditions ^1^ in IPD cases and (%) in the Norwegian population aged 5 years or older, by age group, 2009–2017.

	Number (%) of IPD Cases 5 to 49 y	% In the Population 5 to 49 y	Number (%) of IPD Cases 50 to 64 y	% In the Population 50 to 64 y	Number (%) of IPD Cases 65+ y	% In the Population 65+ y
Population size	1012 (18)	63.8	1406 (25)	19.5	3117 (56)	16.7
Male sex	569 (56)	51.3	690 (49) *	51.8	1536 (49)	47.3
Female sex	443 (44)	48.7	714 (51) *	49.2	1581 (51)	55.3
**Medical Risk Groups ^1^**						
No risk conditions ^a^	784 (78)	95.96	836 (59)	84.95	1513 (49)	67.02
Medium risk conditions ^b^	104(10)	3.50	316 (23)	12.82	955 (31)	26.42
High risk conditions ^c^	124 (12)	0.54	254 (18)	2.22	649 (21)	6.56
Any risk condition	228 (23)	4.15	570 (41)	16.00	1604 (51)	36.40
One risk condition	147 (15)	3.14	300 (21)	10.46	820 (26)	20.87
Two risk conditions	63 (6)	0.66	189 (13)	2.98	498 (16)	7.54
≥2 risk conditions	18 (2)	0.35	81 (6)	2.56	286 (9)	7.99
**Medical Risk Conditions ^1^**						
Immunodeficiency, HIV disease or asplenia	24 (2)	0.20	19 (1)	0.36	25 (1)	0.30
Chronic kidney disease	25 (2)	0.10	39 ((3)	0.50	222 (7)	2.98
Haematological cancer	52 (5)	0.10	127 (9)	0.43	287 (9)	1.10
Generalised malignancy	23 (2)	0.09	96 (7)	0.85	190 (6)	2.37
Other malignancies	47 (5)	0.61	171 (12)	4.58	549 (18)	12.88
Solid organ transplantation	43 (4)	0.08	32 (2)	0.28	27 (1)	0.29
CSF leakage or cochlear implant	5 (0,5)	0.01	0	0.01	2 (0.1)	0.01
Chronic heart disease	10 (1)	0.24	92 (7)	3.26	506 (16)	11.40
Chronic lung disease	32 (3)	1.62	151 (11)	3.03	551 (18)	7.07
Diabetes mellitus	35 (3)	1.10	128 (9)	5.30	338 (11)	10.05
Alcoholism	24 (2)	0.25	61 (4)	0.62	36 (1)	0.50
Chronic liver disease	12 (1)	0.08	33 (2)	0.31	34 (1)	0.43

^1^ ICD-10 codes from the Norwegian Patient Registry that are included in the medical risk groups are presented in Appendix A. ^a^ No risk conditions means the case had none of the listed risk conditions registered. ^b^ Medium risk conditions include CSF leakage, cochlear implant, chronic heart disease, chronic lung disease, diabetes mellitus, chronic liver disease, other malignancies, alcoholism and without any of high-risk conditions. ^c^ High-risk conditions (i.e., immunosuppressive conditions) include immunodeficiency, HIV, asplenia, chronic kidney disease, nephrotic syndrome, leukaemia, lymphoma, generalized malignancies or transplantation. * Two cases in the age group 50 to 64 y had missing information about sex. The no risk, medium risk and high-risk groups are mutually exclusive. The medical risk conditions are not mutually exclusive.

**Table 2 microorganisms-09-01774-t002:** Risk ratios between medical risk groups and those with no medical risk by risk strata, age and vaccine group in Norway, 2015 to 2017 (n = 1622).

Risk Group/Condition	Risk Ratios (95% CI)	
	5 to 64 Years (n = 602)	65+ Years (n = 1020)
All types of IPD, n = 1622		
Medical Risk Groups		
Medium risk ^II^	4.9 (4.0 to 6.0)	1.8 (1.5 to 2.0)
High risk ^III^	25.5 (20.6 to 31.7)	4.2 (3.6 to 5.0)
One comorbidity	5.7 (4.6 to 7.1)	2.0 (1.7 to 2.3)
Two comorbidities	16.0 (12.6 to 20.3)	3.0 (2.5 to 3.6)
≥Two comorbidities	6.6 (4.4 to 9.9)	1.7 (1.4 to 2.1)
Medical Risk Conditions		
Immunodeficiency, HIV-disease and asplenia	11.9 (6.8 to 20.6)	4.1 (2.2 to 7.6)
Chronic kidney disease	13.9 (8.2 to 23.7)	2.8 (2.2 to 3.6)
Haematological cancer	68.1 (50.7 to 91.4)	10.1 (8.1 to 12.7)
Generalised malignancies	32.4 (23.0 to 45.5)	4.2 (3.3 to 5.4)
Solid organ transplants	36.8 (23.7 to 57.1)	6.8 (4.2 to 11.0)
Chronic heart disease	6.2 (4.1 to 9.4)	2.3 (1.9 to 2.7)
Chronic lung disease	6.1 (4.5 to 8.3)	3.6 (3.0 to 4.3)
Diabetes mellitus	6.5 (4.9 to 8.6)	1.6 (1.3 to 2.0)
Alcoholism	17.7 (11.9 to 26.3)	2.8 (1.6 to 5.1)
Chronic liver disease	21.2 (12.2 to 36.8)	3.7 (2.1 to 6.5)
PCV13 IPD, n = 398		
Medium risk ^II^	3.5 (2.3 to 5.2)	1.2 (0.9 to 1.7)
High risk ^III^	16.0 (10.3 to 24.9)	3.2 (2.2 to 4.6)
One comorbidity	4.0 (2.7 to 6.0)	1.4 (1.0 to 1.9)
Two comorbidities	9.8 (6.0 to 15.9)	2.5 (1.7 to 3.6)
≥Two comorbidities	4.6 (2.0 to 10.3)	1.1 (0.6 to 1.8)
non-PCV13 IPD, n = 1211		
Medium risk ^II^	5.0 (3.9 to 6.4)	2.0 (1.7 to 2.3)
High risk ^III^	29.6 (23.3 to 37.7)	4.5 (3.7 to 5.4)
One comorbidity	6.1 (4.8 to 7.8)	2.2 (1.9 to 2.6)
Two comorbidities	17.4 (13.3 to 22.8)	3.2 (2.6 to 3.9)
≥Two comorbidities	7.1 (4.5 to 11.4)	1.9 (1.5 to 2.4)
PPV23 IPD, n = 1132		
Medium risk ^II^	4.3 (3.4 to 5.5)	1.6 (1.3 to 1.9)
High risk ^III^	19.7 (15.2 to 25.6)	3.4 (2.8 to 4.2)
One comorbidity	4.9 (3.8 to 6.2)	1.8 (1.5 to 2.2)
Two comorbidities	12.8 (9.7 to 17.0)	2.6 (2.1 to 3.3)
≥Two comorbidities	5.1 (3.1 to 8.5)	1.2 (0.9 to 1.9)

The numbers are corrected for missing serotypes (n = 29) proportional to the distribution of vaccine types among those with available serotype information (n = 1593). ^II^ Medium risk conditions include CSF leakage, cochlear implant, chronic heart disease, chronic lung disease, diabetes mellitus, chronic liver disease, other malignancies, alcoholism and without any high-risk conditions. ^III^ High-risk conditions (i.e., immunosuppressive conditions) include immunodeficiency, HIV, asplenia, chronic kidney disease, nephrotic syndrome, leukaemia, lymphoma, generalised malignancies or transplantation.

**Table 3 microorganisms-09-01774-t003:** Invasive pneumococcal disease in the population ≥5 years by vaccine serotype and medical risk group, Norway 2015–2017.

Risk Group	N	PCV13	PCV15	PCV20	PPV23
All IPD	1593	24%	43%	59%	70%
No risk	841	29%	49%	66%	76%
Medium risk	438	19%	40%	55%	66%
High risk	314	19%	31%	46%	59%
One comorbidity	377	19%	40%	56%	66%
Two comorbidities	249	20%	35%	47%	63%
≥Two comorbidities	126	17%	28%	43%	54%

In 2016 to 2017, 29 (2%) of the 1622 IPD cases had missing information on serotype and are excluded from this table. PCV13 covers serotypes 4, 6B, 9V, 14, 18C, 19F, 23F, 1, 3, 5, 6A, 7F and 19A. PCV15 covers PCV13 serotypes + 22F and 33F. PCV20 covers PCV15 serotypes and 8, 10A, 11A, 12F, and 15B. PPPV23 serotypes include PCV13 minus 6A, and 2, 8, 9N, 10A, 11A, 12F, 15B, 17F, 20, 22F and 33F.

## Data Availability

The data presented in this study are available on request from the corresponding author. The data are not publicly available due to regulations in the Norwegian Health Research Act and the Norwegian Data Protection Act for use (and storage) of Personal Data related to health. To receive access to the data the applicant will need to provide an ethical approval from their IRB or equivalent body, and an exemption from the duty of confidentiality from Health Registry controllers in Norway.

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
