# Peer review of "The Risk of Invasive Pneumococcal Disease Differs between Risk Groups in Norway Following Widespread Use of the 13-Valent Pneumococcal Vaccine in Children"

_microorganisms, 2021, doi:10.3390/microorganisms9081774_

Round 1

Reviewer 1 Report

The article by Winje et al. is a retrospective study examining the risk of IPD in different age groups in relation to vaccine status and type as well as underlying conditions that are present in the patients. Using extensive surveillance data collected by various Norwegian surveillance systems an extremely comprehensive view is given of IPD rates and risk factors in a Norwegian setting. Overall the paper is extremely well written and presents the data in understandable figures and tables to support the findings and claims of the articles. There is very little that needs to be addressed for publication and this article will add important information into vaccine release and scheduling when higher valent vaccines are available. A few comments are included below, but the article as is is very compelling.

Figure 3. Add extra information in the legend about what the solid and dashed lines represent. Mentioned below the figure but the information in the legend will also be useful.

Page 6 and 7 text in reference to the risk groups of figure 2 and 3. The explanation of what the risk conditions are repeated on page 7, this seems unnecessary and just referencing the risk conditions as described above should be sufficient.  

Author Response

Thank you for the positive response to our manuscript and for your kind  suggestions to improve it.

Figure 3 – we will add information in the legend about the solid and dashed lines. The updated figures will be submitted later due to waiting time at the infographics department at NIPH. We have agreed with the editorial team that we may submit the updated figures at a later time. 

Page 6 and 7 We included information about the risk conditions so that the information in the table could stand alone. We have now taken out the information about the risk conditions from the figures.

Reviewer 2 Report

Figure 1. The image resolution is poor and the results need to be standardized. The axes need to be presented in the same way, same font, same color.

Figure 2. This figure has the same issues as the first one.

Figure 3. In this figure, the authors add an issue in the numbers on the y axis. Please check the decimal place. Do you want the dot or the comma to indicate them? Standardized this point

Table 3. need to be standardized

Author Response

Thank you for the positive response to our manuscript and for your kind suggestions to improve it. 

The figures will be updated, but are yet not available due to waiting time at the inforgraphics department at NIPH. We have agreed with the editorial team that we may submit these at a later time. 

Response to comments:

The image resolution in figures 1-3 will be improved. 

The figures will be standardized in terms of same font and same color. However, we suggest to keep the scale on the y-axis different for the All-type IPD in figures 1 and 2. Alternatively all axes will have to be scaled according to all-type IPD. This would make the differences between groups more difficult to see for the reader. It is mentioned in the figures footnotes and in the text that the y-axis differ in scale. We believe this is sufficient for the reader to become aware of this. However, if the editors prefer the axis to be scaled similarly we will amend them accordingly.

In Figure 3, we will change to punctuation to indicate decimals.  We have also added a line to make the reader aware that the y-axis starts on 0.6. 

Table 3 is now standardized.  

Reviewer 3 Report

The paper deals with an essential aspect of vaccination, thus assessing the validity of vaccination against IPD, taking into account the risk of contracting the disease later. The manuscript is well prepared, with few punctuation or spelling errors (eg, KEYWORDS: it is Streptococcus pneumonia, but it should be Streptococcus pneumoniae). The font is not the same everywhere (e.g., in Table 2). It requires standardization. Of the 45 cited papers, 24 were published in the last five years; the rest are older, which could be updated. The tables are clear, and the figures present the epidemiological situation over the years (Fig. 2) or the risk of developing IPD taking into account the risk factors. The great advantage of the study is the vast population group (n = 5535 cases of IPD) and the comprehensive collection of information on risk factors. Even in the case of serotype information missing for 135 cases (2.4%), it does not matter much considering such a large study group. The study was well planned, albeit retrospective. The authors indicate how important is the prevention of infections, especially among people at medical risk conditions.

Author Response

We thank the reviewer for the positive response to our manuscript and for your kind suggestions to improve it. 

We have corrected the spelling errors and the font is now the same throughout the manuscript.

The reviewer commented that many of the references were published more than 5 years ago, and suggested that we update the references with more recent publications. We have gone through allreferences published before 2016: 

  • We suggest to keep three references (1-3) even if they were published several years ago; (i) they cover similar topics as in the current manuscript, (ii) few recent publications cover the same topic with similar detail and (iii) they are important for the discussion in the current manuscript.
  • We took out nine citations without replacing them. The statements in the manuscript are already sufficiently covered by other more recent references. The earlier publications were included to provide a full picture.
  • We replaced four citations with more recent references
  • We added one recent reference which was not previously included.

We hope this is acceptable to the reviewer.

An overview is provided below. 

  1. Pilishvili T, Zell ER, Farley MM, Schaffner W, Lynfield R, Nyquist AC, et al. Risk factors for invasive pneumococcal disease in children in the era of conjugate vaccine use. Pediatrics. 2010;126(1):e9-17. Taken out – other references sufficient
  2. Muhammad RD, Oza-Frank R, Zell E, Link-Gelles R, Narayan KM, Schaffner W, et al. Epidemiology of invasive pneumococcal disease among high-risk adults since the introduction of pneumococcal conjugate vaccine for children. Clin Infect Dis. 2012. Kept – same topic, important for discussion
  3. van Deursen AMM, van Mens SP, Sanders EA, Vlaminckx BJ, De Melker HE, Schouls LM, et al. Invasive pneumococcal disease and 7-valent pneumococcal conjugate vaccine, the Netherlands. Emerg Infect Dis. 2012;18(11). Kept – same topic, important for discussion
  4. van Hoek AJ, Andrews N, Waight PA, Stowe J, Gates P, George R, et al. The effect of underlying clinical conditions on the risk of developing invasive pneumococcal disease among hospitalised patients in England. J Infect. 2012;65(1):17-24. Kept – same topic, important for discussion
  5. Iroh Tam PY, Madoff LC, Coombes B, Pelton SI. Invasive pneumococcal disease after implementation of 13-valent conjugate vaccine. Pediatrics. 2014;134(2):210-7. Taken out – other references sufficient
  6. Ricketson LJ, Nettel-Aguirre A, Vanderkooi OG, Laupland KB, Kellner JD. Factors influencing early and late mortality in adults with invasive pneumococcal disease in Calgary, Canada: a prospective surveillance study. PLoS One. 2013;8(10):e71924. Taken out – replaced by: Chen HM, H. Horita, N., Hara YK, N. Kaneko, T. Prognostic factors for mortality in invasive pneumococcal disease in adult: a system review and meta-analysis. 2021.
  7. Neralla S, Meyer KC. Drug treatment of pneumococcal pneumonia in the elderly. Drugs Aging. 2004;21(13):851-64. Taken out – replaced by: Chen HM, H. Horita, N., Hara YK, N. Kaneko, T. Prognostic factors for mortality in invasive pneumococcal disease in adult: a system review and meta-analysis. 2021.
  8. Kyaw MH, Rose CE, Jr., Fry AM, Singleton JA, Moore Z, Zell ER, et al. The influence of chronic illnesses on the incidence of invasive pneumococcal disease in adults. J Infect Dis. 2005;192(3):377-86. Taken out – other references sufficient
  9. Klugman KP. A tale of 2 pneumococcal vaccines. Clin Infect Dis. 2014;58(7):925-7. Taken out – replaced by: Winje BA BJ, Vestrheim DF, Denison E, Lepp T, Roth A, Valentiner-Branth P, Slotved HC, Storsaeter J,. Efficacy and effectiveness of pneumococcal vaccination in elderly - an update of the literature. Oslo, Norway: Norwegian Institute of Public Health; 2019 Contract No.: ISBN digital: 978-82-8406-053-8.
  10. Vestrheim DF, Hoiby EA, Bergsaker MR, Ronning K, Aaberge IS, Caugant DA. Indirect effect of conjugate pneumococcal vaccination in a 2+1 dose schedule. Vaccine. 2010;28(10):2214-21. Taken out, citation not necessary. We also deleted the following sentence: Details on microbiological surveillance in Norway have been described before.
  11. Bakken IJ, Suren P, Haberg SE, Cappelen I, Stoltenberg C. [The Norwegian patient register--an important source for research]. Tidsskr Nor Laegeforen. 2014;134(1):12-3. Taken out – replaced by: Bakken IJA, A. M. S., Knudsen, P.  Johansen, K. I. Vollset, S. E. . The Norwegian Patient Registry and the Norwegian Registry for Primary Health Care: Research potential of two nationwide health-care registries. Scand J Public Health,. 2020;48(1):49-55.
  12. Lepoutre A, Varon E, Georges S, Dorleans F, Janoir C, Gutmann L, et al. Impact of the pneumococcal conjugate vaccines on invasive pneumococcal disease in France, 2001-2012. Vaccine. 2015;33(2):359-66. Taken out, the citation from Thorax (Hanquet 2018) is sufficient
  13. D'Ancona F, Caporali MG, Del Manso M, Giambi C, Camilli R, D'Ambrosio F, et al. Invasive pneumococcal disease in children and adults in seven Italian regions after the introduction of the conjugate vaccine, 2008-2014. Epidemiol Prev. 2015;39(4 Suppl 1):134-8. Taken out, the citation from Thorax (Hanquet 2018) is sufficient
  14. von Gottberg A, de Gouveia L, Tempia S, Quan V, Meiring S, von Mollendorf C, et al. Effects of vaccination on invasive pneumococcal disease in South Africa. N Engl J Med. 2014;371(20):1889-99. Taken out – other references sufficient
  15. Lujan M, Burgos J, Gallego M, Falco V, Bermudo G, Planes A, et al. Effects of immunocompromise and comorbidities on pneumococcal serotypes causing invasive respiratory infection in adults: implications for vaccine strategies. Clin Infect Dis. 2013;57(12):1722-30. Taken out – other references sufficient
  16. Klemets P, Lyytikainen O, Ruutu P, Ollgren J, Nuorti JP. Invasive pneumococcal infections among persons with and without underlying medical conditions: implications for prevention strategies. BMC Infect Dis. 2008;8:96. Taken out – other references sufficient

An additional recently published review on the risk associated with age is added as a reference:

  1. Grant LR, Slack MPE, Yan Q, Trzcinski K, Barratt J, Sobczyk E, et al. The epidemiologic and biologic basis for classifying older age as a high-risk, immunocompromising condition for pneumococcal vaccine policy. Expert Rev Vaccines. 2021:1-15.